# Maternal Obesity and the Uterine Immune Cell Landscape: The Shaping Role of Inflammation

**DOI:** 10.3390/ijms21113776

**Published:** 2020-05-27

**Authors:** Lauren E. St-Germain, Barbara Castellana, Jennet Baltayeva, Alexander G. Beristain

**Affiliations:** 1The British Columbia Children’s Hospital Research Institute, Vancouver, BC V5Z 4H4, Canada; lauren.stgermain@bcchr.ca (L.E.S.-G.); bcastellana@bcchr.ca (B.C.); jbaltayeva@bcchr.ca (J.B.); 2Department of Obstetrics & Gynecology, The University of British Columbia, Vancouver, BC V6Z 2K8, Canada

**Keywords:** maternal obesity, immune cells, inflammation, pregnancy, placentation

## Abstract

Inflammation is often equated to the physiological response to injury or infection. Inflammatory responses defined by cytokine storms control cellular mechanisms that can either resolve quickly (i.e., acute inflammation) or remain prolonged and unabated (i.e., chronic inflammation). Perhaps less well-appreciated is the importance of inflammatory processes central to healthy pregnancy, including implantation, early stages of placentation, and parturition. Pregnancy juxtaposed with disease can lead to the perpetuation of aberrant inflammation that likely contributes to or potentiates maternal morbidity and poor fetal outcome. Maternal obesity, a prevalent condition within women of reproductive age, associates with increased risk of developing multiple pregnancy disorders. Importantly, chronic low-grade inflammation is thought to underlie the development of obesity-related obstetric and perinatal complications. While diverse subsets of uterine immune cells play central roles in initiating and maintaining healthy pregnancy, uterine leukocyte dysfunction as a result of maternal obesity may underpin the development of pregnancy disorders. In this review we discuss the current knowledge related to the impact of maternal obesity and obesity-associated inflammation on uterine immune cell function, utero-placental establishment, and pregnancy health.

## 1. Inflammation and Pregnancy

Inflammation is an essential part of healthy pregnancy. Inflammatory factors, produced by diverse cell subtypes within the maternal-fetal interface, coordinate critical events in embryo implantation [1], placental development [2], and parturition [3]. Nonetheless, the consequences of infection-induced inflammation in pregnancy are perhaps more widely appreciated and studied due to outcomes such as severe preterm birth [4]. Common to both physiological and pathophysiological forms of inflammation is the production of inflammatory cytokines that impact systemic and local immune cell responses [5]. Importantly, the amplitude, duration, and composition of inflammatory signals dictate how biological processes in pregnancy are ultimately impacted.

The first 4–7 weeks in human pregnancy, when the placenta is immature and maternal blood-flow within the intervillous space has not yet been established, define a critical period where developmental trajectories and environmental inputs impose significant influence in shaping pregnancy success. The uterine environment during this developmental window is defined by the predominance of pro-inflammatory factors, including cytokines and chemokines, produced by diverse cell types of the placenta and uterine stroma [5,6,7]. These factors function in part to recruit immune cells from the periphery to participate in uterine tissue remodeling and fine-tuning immune tolerance towards the developing fetus and placenta [6,8,9]. Following the first trimester, the uterine pro-inflammatory environment transitions into one that is more regulatory by mid pregnancy, and this is maintained until the initiation of labor [8].

Uterine immune cell processes are shaped by their cellular environment. Alterations in cytokine gradients and the balance in production of pro-inflammatory and regulatory factors play central roles in coordinating biological processes important in pregnancy health. Deviations in inflammatory factor composition leading to aberrant inflammation likely impact important uterine immune cell processes and may precipitate the development of pregnancy-related disorders such as miscarriage, preterm birth, and even subtypes of preeclampsia [4]. This review aims to concisely describe the overall decidual immune cell landscape in pregnancy. Importantly, this review also sets out to outline how persistent chronic inflammation, especially the type of inflammation associated with obesity, impacts local uterine immune cell function and potentially contributes to the development, or even initiation, of adverse pregnancy outcomes.

### The Uterine Immune Cell Landscape

The uterus is a dynamic organ that is subjected to cyclic restructuring during the menstrual cycle and undergoes enormous transformations during pregnancy. One of these transformations, termed decidualization, is the transformation of the endometrial lining of the uterus into secretory and hypertrophic/rounded cells to establish a microenvironment favorable for pregnancy and to provide physical support for placental attachment [10,11]. The decidua becomes the primary site of interaction and intercommunication between placental-derived trophoblasts and maternal-derived stroma. Decidual stromal cells represent the main cellular component of the gravid uterus and contribute to the development of pregnancy by fostering complex communication with trophoblasts and orchestrating behavioral changes in decidual immune cells via secretion of cytokines and chemokines [1,2,3]. Amongst these secreted factors are prolactin (PRL), insulin-like growth factor binding protein (IGFBP)-1, vascular endothelial growth factor (VEGF), transforming growth factor (TGF)-β1, and interleukin (IL)-15; together, these factors regulate angiogenesis, promote trophoblast invasion, and control uterine natural killer cell (uNK) survival [11,12,13,14,15].

At the maternal-fetal interface, a diverse make-up of decidual leukocytes exists: uNKs, macrophage(s) (Mφ), and T cells are the most abundant subtypes (Figure 1). However, populations of dendritic cells (DCs), innate lymphoid cells (ILCs), and neutrophils also reside within the maternal-fetal interface and likely play important roles in pregnancy [16,17,18,19,20]. Studies in humans and mice show that uterine immune cells are phenotypically and behaviorally different from their peripheral counterparts, most likely as a result of decidual factor influence and their state of tissue residency. However, many decidual leukocytes are actively recruited from the periphery to the placental-maternal interface by chemoattractants secreted by trophoblasts and decidual stromal cells [21,22]. It is estimated that up to 40% of decidual cellular content in the first trimester are leukocytes, a fact clearly highlighting the inflammatory nature of pregnancy [17].

*Uterine natural killer cells* are amongst the most thoroughly studied uterine immune cells due in large part to their abundance in human and mouse decidua. They are most abundant in the first trimester, constituting ~70% of uterine immune cells in humans [17] and ~35% of uterine immune cells in mice [24]. In both humans and mice, uNK numbers peak around the time that uterine arterial remodeling is initiated, and following this, gradually decline in numbers as pregnancy progresses (Figure 1) [24,26]. In contrast to peripheral NKs that harbor efficient innate sentinel functions, uNKs are not normally cytotoxic but are instead major producers of cytokines, chemokines, and angiokines [27,28]. In healthy pregnancies, uNKs localize to invading trophoblasts [29] and uterine spiral arteries [30], suggesting that uNKs may regulate trophoblast biology and/or spiral artery remodeling. Indeed, uNK-secreted factors have been shown to both promote as well as restrict extravillous trophoblast (EVT) motility via hepatocyte growth factor, IL-8, C-X-C motif chemokine (CXCL)-10, and interferon (IFN)-γ secretion [27,31,32]. However, the intricacies of uNK-trophoblast interactions must be interpreted with caution as most studies investigating this phenomenon have resorted to using trophoblast cell lines and uNK derived supernatants instead of uNK/EVT primary cell co-cultures. Moreover, because uNKs produce tumor necrosis factor (TNF)-α, placental growth factor (PlGF), VEGF-C, and matrix metalloproteinases (MMPs), a major biological function ascribed to uNKs relates to their importance in uterine spiral artery remodeling [2,28,33]. Indeed, in rodents, uNK-deficiency results in dampened vascular density and impaired remodeling of spiral arteries [34,35]. While a similar role for uNKs in humans has not explicitly been shown, uNK spatial localization and the detailed characterization of uNK-derived secreted factors suggests that uNKs in human pregnancies perform similar uterine-vascular remodeling tasks as documented in mice.

*Macrophages* are the second most abundant leukocyte within the maternal-fetal interface with frequencies between ~20%–30% of total immune cells [16,36]. Mφ are highly plastic cells that adopt a broad range of inflammatory characteristics defined in part by the factors they secrete [37]. To this end, polarized states of Mφ can be described as pro-inflammatory (i.e., M1-like) and regulatory (i.e., M2-like), but it is crucial to appreciate that the majority of Mφ fall within a spectrum of these two extremes [38]. In fact, Mφ often express surface markers and secrete factors that are suggestive of a mixed M1/M2-like phenotype [39,40]. In the decidua of healthy pregnancies, Mφ are believed to be skewed towards a homeostatic or regulatory anti-inflammatory M2-like state [41] that is initiated and maintained by the secretion of macrophage colony-stimulating factor (M-CSF) and IL-10 by trophoblasts and decidual stromal cells [42]. Similar to uNKs, Mφ aid in spiral artery remodeling via the secretion of MMPs and angiogenic growth factors VEGF-A, angiopoietin (Ang)-1, and Ang-2 [41,43]. Through phagocytic processes, decidual Mφ additionally aid in the “cleanup” or removal of apoptotic cells and debris that accumulate within the placental-maternal interface as a result of tissue remodeling, growth, and differentiation [44,45]. Lastly, decidual Mφ likely modulate placental development in part by secreting factors known to affect trophoblast biology. For example, Mφ-derived IL-8, TNF-α, and IL-10 alter trophoblast migration; however, conflicting evidence exists as to whether the combined effect of these factors is pro- or anti-migratory [46,47,48,49].

*Dendritic cells* are closely related to Mφ but are comparatively more potent in antigen capturing (immature DCs) and presentation (mature DCs) [50]. DCs play an important role in T cell expansion and polarization through antigen specific immune responses, and thus, function to bridge the innate and adaptive immune systems [51]. Decidual DCs are present at much lower frequencies than other immune cell types, comprising about 1% of total uterine leukocytes [52,53]. Immature DC proportions are generally higher than mature DC frequencies, most likely due to the overall effect of decidual stromal cells in impeding DC differentiation and maturation [54]. Thus, the majority of decidual DCs are maintained in a resting immature state exhibiting tolerogenic effects as supported by in vitro findings suggesting involvement of DCs in skewing immune responses towards a regulatory T helper type (Th) 2 [55]. Despite their low numbers within the maternal-fetal interface, the importance of DCs for maintenance of pregnancy cannot be overlooked. In mice, DC ablation results in embryo resorption and impaired decidual angiogenesis, emphasizing the importance of DCs in pregnancy [56,57]. Notably, DCs participate in a close cooperative dialogue with other decidual leukocytes [53,58]. In a reciprocal manner, DCs enhance proliferation and differentiation of uNKs [59], whereas interaction of uNKs with DCs improves the ability of DCs to induce immunosuppressive Treg expansion and activity [60].

*T cells* comprise 10%–20% of decidual leukocytes, of which 30%–45% are cluster of differentiation (CD)4^+^ helper cells and 45%–75% are effector/cytotoxic CD8^+^ cells [17,61]. While decidual CD8^+^ T cell cytotoxic activity has been demonstrated through in vitro experimentation [62], effector T cell interaction with human leukocyte antigen (HLA)-C expressed on trophoblasts drives the expression of co-inhibitory molecules on CD8^+^ cells that dampen their cytotoxic responses [63,64]. Th1, Th2, and Th17 cells constitute the main subtypes of effector CD4^+^ T cells [65]. Th1 secrete IFN-γ and IL-2 as their signature cytokines and these factors in part orchestrate the complex interactions and cross-talk between decidual immune and endometrial cells with the fetal semiallograft, which contribute to pregnancy pathologies such as recurrent spontaneous abortion [66,67,68,69].

Th2, on the other hand, are producers of immunomodulatory cytokines IL-4, IL-13, and IL-10, which act to minimize Th1 elicited responses [70,71,72]. Until recently, the prevalence of Th2-type suppressive immune responses over Th1 responses in pregnancy was used to explain maternal tolerance towards the semi-allogeneic fetus [73]. This concept was supported by studies that observed a predominance of Th1-type responses in cases of recurrent spontaneous abortion and preeclampsia [74,75,76,77]. However, the Th1/Th2 paradigm has been proven overly simplistic and incomplete to fully account for the mechanism of protection of the fetus from maternal immune cell attack. Instead, the Th1/Th2 paradigm has now been expanded to include Th17 and Tregs as contributors to maternal-fetal tolerance [78]. Th17 have only recently been characterized in the uterus and produce IL-17 and IL-22 [79]. In pregnancy, there is evidence for involvement of Th17 cells in recurrent spontaneous abortion and pre-eclampsia [80,81,82]. Tregs (predominately CD4^+^CD25^+^FoxP3^+^ Tregs) dampen pro-inflammatory functions of other T cells, primarily via secretion of anti-inflammatory cytokines such as IL-10 and TGFβ [83,84]. Tregs also act to recognize fetal antigens with the help of maternal antigen-presenting cells and induce tolerance towards the allogeneic fetus [63,85,86]. Thus, a delicate balance exists between Th1, Th2, Th17, and Tregs at the maternal fetal interface to ensure tolerance and a successful pregnancy.

*Neutrophils* are classically short-lived granulocytes with antimicrobial effector functions and are amongst the first leukocytes to be recruited to sites of infection [87]. Neutrophils are present in the first trimester decidua and produce heparin binding-epidermal growth factor (HB-EGF), a factor that is important in neo-angiogenesis and blastocyst implantation [88]. Furthermore, neutrophil exposure to pregnancy hormones (estrogen and progesterone) induces a unique population of Tregs displaying pro-angiogenic properties [89]. However, compared to other immune cell types within the uterus, the importance of neutrophils in pregnancy is less well understood in part because of immune cell isolation approaches utilizing density gradients that exclude granulocyte enrichment.

*Innate lymphoid cells* are a heterogenous group of cells that exist in mucosal and barrier tissues and lack antigen-specific recognition receptors [90] and are categorized as ILC1s, ILC2s, and ILC3s based on their phenotype. ILC1s secrete IFN-γ and include natural killer cell (NK)s and other non-cytotoxic ILC1s. ILC2s are dependent on the transcription factors GATA-binding protein 3 (GATA3) and retinoic acid receptor-related orphan receptor (ROR)-α for their development, while ILC3s require the transcription factor RORα for their development and secrete IL-17 and/or IL-22 [91]. ILCs exist in the pregnant uterus of both humans and mice [20,92,93]. ILC1 and ILC3 frequencies are highest in the first trimester, whereas ILC2s are present in greater proportions in the third trimester of pregnancy [94]. ILC3s are the most extensively studied of human decidual ILCs. These cells produce granulocyte-macrophage colony stimulating factor (GM-CSF) and IL-8, which were found to aid the migration and survival of neutrophils in first trimester decidua [88]. A recent study identified five unique subsets of decidual ILCs in human pregnancy: proliferating NKs (dNKp), decidual/uterine NKs (dNK) 1–3, and decidual ILCs (dILC3s) [95]. Within these ILC variants, significant heterogeneity in cytokine/chemokine secretion and receptor expression patterns exists. dNK1 express NK receptors (NKR) like killer-cell immunoglobulin-like receptors (KIR) that bind HLA class I ligands expressed on EVT, while dNK2 and dNK3 are devoid of NKRs. Where dNK1 control uterine vascular remodeling, dNK2 and dNK3 are major producers of chemokines that likely control recruitment of dendritic cells and modify EVT function. Hence, dILCs are emerging as a unique and specialized group of cells with potential to contribute to pregnancy success.

## 2. Obesity-Driven Inflammation

Subcutaneous adipose tissue plays a central role in energy homeostasis and is a major site for body energy stores in the form of “fat” or triglycerides [96,97]. In addition to functioning as an energy reservoir, adipose tissue also senses energy demands and produces factors such as adipokines and cytokines to regulate other metabolic tissues [96,97]. Adipose tissue of healthy weight people is a heterogeneous composite of adipocytes, multiple subtypes of immune cells, and structural connective tissue [96,97]. The leukocyte composition of healthy adipose tissue is characterized by a predominant type-II/regulatory phenotype defined in part by the prevalence of M2-like Mφ, Th2 T cells, and Tregs [98]. However, immune cell subsets appearing at lower frequencies (i.e., eosinophils, ILC2s, DCs, γδ T cells, and natural killer T cells) also reside within fat tissue, and together coordinate metabolic homeostasis within both subcutaneous adipose reservoirs and metabolic active sites such as the liver and muscle [98].

In obesity, over-nutrition leads to the accumulation of fat within mature adipocytes, resulting in cellular hypertrophy and the ectopic distribution of fat stores to visceral locations within the body [96]. The increase in adipocyte fat as well as metabolic by-products of triglycerides such as glycerol and free fatty acids (FFA) results in lipotoxicity and increased rates of adipocyte death [97]. Together, these processes trigger an inflammatory cascade defined by the infiltration of peripheral immune cells into adipose tissue (predominately monocytes that transform into pro-inflammatory M1-like Mφ), and the heightened and chronic production of pro-inflammatory cytokines [99]. It is hypothesized that this is the initiating event resulting in downstream obesity-related aberrant inflammation [100,101]. Subsequently, adipose tissue dysfunction leads to insulin resistance in insulin-sensing tissues like liver and muscle, and the combined stress of pancreatic β cell production of insulin and exposure to systemic inflammatory cytokines of adipose tissue origin perpetuates this viscous cycle [100,101]. Notably, the chronic production of pro-inflammatory factors derived from adipocytes [monocyte chemoattractant-protein-1 (MCP-1), IL6] and adipose tissue Mφ (TNF-α, IL1-β) impacts distal sites within the body leading to endothelial dysfunction, and contributes to processes underlying cardiovascular disease, metabolic diseases, and hypertension [100,101].

*Obesity-associated immune cell changes*. Chronic inflammation within adipose tissue disrupts immune homeostasis by promoting and perpetuating alterations in the biology of immune cells [101]. T cells are thought to be a major inducer of inflammation in obesity through secretion of proinflammatory mediators prior to monocyte infiltration and macrophage differentiation [102]. On one hand, monocyte accumulation, hyperplasia, and apoptosis promote the recruitment of neutrophils and T cells to the adipose tissue [103]. On the other hand, changes in the adipose inflammatory factor make-up instruct resident immune cells to undergo a series of phenotypic changes that further promote and sustain local inflammation via positive feed-back loops [103]. Adipose tissue resident Mφ and DCs polarize towards unique pro-inflammatory phenotypes [99] that perpetuate the shift of CD4^+^ T cells from Th2 to Th1 characteristics and transform anti-inflammatory Tregs to pro-inflammatory Th17 cells [104]. Overall, obesity is characterized by both local and systemic types of immune cell inflammation. How/if adipose-derived inflammation resulting from obesity impacts cellular processes within the maternal-fetal interface is currently not well understood but is an active area of research and an attractive paradigm linking maternal obesity to heightened risks of multiple pregnancy disorders.

## 3. Maternal Obesity

Obesity is a serious and rising cause of obstetrical and perinatal morbidity. In North America, over 20% of women of childbearing age are considered obese (body mass index (BMI) >30 kg/m^2^) [105]. Obese women are about three times more likely to develop pregnancy complications, including gestational diabetes, miscarriage, superfertility, stillbirths, and preeclampsia [106,107,108,109]. Obesity may also associate with increased risk of preterm birth, especially extreme preterm birth, which together are responsible for ~3 million neonatal deaths annually worldwide [110,111]. However, the underlying biological processes in maternal obesity driving adverse events in pregnancy are poorly understood but may involve interactions of pregnancy-related processes with obesity-associated low-grade chronic inflammation. These interaction between obesity, pregnancy, and pregnancy-specific disorders are summarized in Figure 2. The generation of reliable rodent models that mimic obesity in pregnancy is crucial to understand the mechanisms how obesity associates with pregnancy complications. The current available models for obesity and inflammation in pregnancy are summarized in Table 1.

Obesity-induced metabolic dysfunction and pregnancy complications share mechanistic similarities. Specifically, proinflammatory factors such as HMGB1 [112,113,114,115], TNF-α [116,117], and IL-1β [118,119,120] play critical roles in the inflammation underlying both obesity and preeclampsia. These key inflammatory mediators when upregulated in obesity [102] may potentiate aberrant maternal inflammation that in turn contribute to the development of pregnancy disorders [121]. Chronic inflammatory conditions associate with both early [122] and late-onset preeclampsia [123], where late-onset preeclampsia, thought to be initiated by maternal factors such as altered metabolism and endothelial dysfunction, is more prevalent in obese women [123]. This association might be further exacerbated in obese mothers who are primigravid [124] or in women presenting with an additional pre-existing chronic inflammatory condition [125]. Obesity alone associates with physiological changes such as vascular and endocrine alterations, processes that are also affected in preeclampsia [126]. Therefore, it is likely that pre-existing inflammatory conditions, such as obesity, prime biological processes such as immune activity, metabolic function, and cardiovascular health to be further exacerbated by the physiological demands of pregnancy. The following sections will discuss the possible effects of obesity-associated chronic inflammation on placental development and immune cell biology within the maternal-placental interface.

### 3.1. Impact of Maternal Obesity on Placentation

Impaired placentation resulting from obesity may be one of the reasons why obese women are at higher risk of aberrant pregnancy outcomes. Pre-pregnancy BMI positively associates with greater placental weight [174,175], increased birth weight [174,175], and accelerated villus maturation [174]. Moreover, increasing BMI also associates with increased risk of chronic villitis [174,176], characterized by lymphocyte and monocyte infiltration within placental villi. Additionally, placental efficiency, as measured by dividing fetal by placental weight, was reduced in pregnancies of obese women [177]. Obesity also associates with marginal insertion of the umbilical cord and intervillous thrombi in the parenchyma of pregnant women [178].

A global transcriptomic approach designed to generate molecular understandings of how maternal obesity impacts the human placenta identified that placentae of obese women show enrichment of gene signatures related to lipid metabolism, angiogenesis, hormone activity, and cytokine activity [179]. Specifically, levels of VEGF-A and hypoxia-inducible factor (HIF)1-α, factors important in initiating and promoting vasculogenesis and angiogenesis, were reported to be decreased in placentae of obese women [179]. In support of this, histopathological preparations of placentae from pregnancies of obese women showed angiogenic defects [129,180] and fetal growth restriction [181,182,183]. These associations have been corroborated in rodent high-fat diet (HFD) models of maternal obesity, where impairments in overall placental vascularity are also observed [184]. On the maternal side of the placental-uterine interface, maternal obesity in humans and exposure to HFD in mice associate with impaired uterine vascular remodeling defined in part by thicker smooth muscle layers around arteries and narrower arterial conduits [129,130,185,186]. Mouse models of maternal obesity also display poor decidual formation with smaller implantation sites early in pregnancy, but have larger placentae and fetuses at term than control mice, suggesting that fetuses of pregnancies exposed to HFD might compensate later in pregnancy, reaching macrosomia at term [187]. Moreover, defective decidualization in early pregnancy is known to associate with complications later in pregnancy [15].

Angiogenesis and vascular remodeling are pivotal processes in the development of a healthy functioning placental-uterine interface, and thus, obesity-linked impairments in these processes could underlie the biology contributing to adverse pregnancy outcomes. In humans and mice, blood vessel growth and establishment within the placenta and uterus are in part regulated by diverse subsets of cells (i.e., trophoblasts, decidual Mφ, uNKs, and Tregs). It is therefore possible that obesity impacts the biology of diverse subtypes of uterine immune cells in similar ways to how adipose tissue immune cells are altered in obesity.

Obesity-driven increases in circulating lipids, hormones, reactive oxygen species, and cytokines may also impact placental development negatively by affecting the health and function of trophoblasts. For example, low density lipoproteins (LDL) inhibit migration and promote apoptosis of primary EVT in culture [188]. Furthermore, leptin, a pro-inflammatory adipose tissue hormone, when added to primary cultures of human trophoblasts, restrains proliferation and promotes invasion [189], cellular processes linked with mature EVT differentiation, and enhanced uterine arterial remodeling. This supports the finding that pregnant rats exposed to HFD exhibit impaired trophoblast invasion [130].

Studies using rodent models to examine the effect of obesity on placentation, though insightful and useful, do necessitate a level of caution with interpretation of their findings. Important features of pregnancy and placental development that are different between humans and mice include the length of gestation and number of conceptus sites, levels and source of endocrine factors (i.e., progesterone is ovarian in rodents while transitions to placental in humans), and relative developmental trajectories of placental maturation [190,191]. Together, these species-related differences may affect how local and systemic inflammation interact within the maternal-fetal interface. Importantly, differences in placental structure and trophoblast biology are likely also imperative in modifying how the placenta interacts with maternal inflammatory factors. For example, the human placenta is villous and maternal blood bathes chorionic villi within intervillous spaces. By contrast, the rodent labyrinthine placenta establishes an efficient counter-current maternal-fetal blood flow that minimizes hypoxia-reoxygenation stress to cells [190,191]. Human trophoblasts are also more invasive than rodent trophoblasts, and unlike rodent trophoblast giant cells, human EVT are thought to contribute more to uterine artery remodeling [190,192]. How or if such structural and physiological differences affect inflammatory processes (i.e., cytokine half-life, clearance, accessibility) is at this point not well understood. Collectively, human-rodent differences necessitate careful assessment when extrapolating findings from model organisms to human biology.

### 3.2. Obesity-Driven Uterine Immune Cell Dysfunction

Studies have consistently shown that obesity alters immunogenic processes in many types of both peripheral and tissue resident immune cells [193,194,195,196,197,198,199,200,201]. It is plausible that maternal obesity, through the production of pro-inflammatory factors originating from adipose tissue, impacts the uterine immune cell environment. However, the effect of obesity on uterine immune cell processes is essentially unknown, and what is known is limited to a select few studies focusing on uNKs, Mφ, and Tregs that will be summarized below.

*Maternal obesity affects fetal tolerance*. In the uterus, pregnancy coincides with an immunological paradox where maternal immune cells establish tolerance towards foreign antigens and retain immuno-reactive capacity. Recognition of classical (i.e., HLA-C) and non-classical (i.e., HLA-E, HLA-G, HLA-E) MHC-I ligands by uNK, Mφ, DCs, and T cells promotes mechanisms of tolerance [202]. For instance, crosstalk between uNK and trophoblasts that leads to immunotolerance can be initiated via trophoblastic HLA-G and HLA-E interacting with uNK-expressed LILRB1 [203] and NKG2A [204,205] leading to secretion of immuno-suppressive IL-10 and TGF-β [206,207] and dampening of uNK cytotoxicity [204,205,208]. Engagement of uNK-expressed KIR2DL1 to HLA-C expressed by trophoblasts blunts KIR2DS1 activating responses to HLA-C2 [209,210]. Furthermore, HLA-G binding to LILRB2 expressed on decidual APCs promotes a tolerogenic phenotype within decidual Mφ and DCs via secretion of IL-10 and TGF-β; these factors induce Treg expansion and limit T cell effector functions [211,212,213,214]. Alterations in immune function resulting from pro-inflammatory environmental cues could translate into impaired maternal-fetal tolerance resulting in impaired placentation, chronic inflammation, and poor pregnancy outcome [69,213].

Impaired Treg induction within adipose tissue associates with obesity [214]. Therefore, it is possible that maternal obesity impacts uterine Treg populations in a similar manner. However, at this time, only one study has reported the impact of diet induced obesity (DIO) in mice on uterine Tregs [127]. Notably, mice exposed to HFD showed increased frequencies in uterine Tregs than control diet mice, a finding somewhat contrary to what would be expected if maternal obesity potentiates reduced tolerance. Here, the authors suggest that increased frequencies of Tregs might be a form of compensation against a pro-inflammatory environment induced by obesity. Nonetheless, it is plausible that alterations in decidual Treg frequencies may sustain a pro-inflammatory non-tolerogenic environment by enabling uncontrolled T cell effector functions [215] and uNK cytotoxicity [216]. Additionally, Tregs exert some degree of functional plasticity. For example, in the context of pathological conditions such as preeclampsia and type 1 diabetes, Tregs can potentially transform into pro-inflammatory Th17 [217] or Th1 [218] cells. However, whether this transformation occurs in the decidua of obese women is currently unknown.

*Obesity potentiates uterine leukocyte activation.* Obesity dysregulates NK biology [201,219,220,221,222], promoting a highly activated state [193,220,223]. Despite the critical importance of these cells in regulating placental establishment and development, only a few studies have examined the effect of maternal obesity on uNK activity [129,185,224]. There is conflicting evidence emerging from mouse models of DIO examining alterations on NKs within the utero-placental environment. Baltayeva et al. showed that a 13-week HFD exposure led to overall increased uNK activity [129]. Contrary to this observation, Parker et al. showed that NK activity following HFD exposure remained unchanged compared to control diet fed mice [127]. In the latter study, NKs were isolated from draining inguinal lymph nodes instead of isolated directly from uterine mucosa. Thus, it is likely that the NKs analyzed were quite distinct between these two studies, where one reflects a heterogeneous composition of conventional and tissue resident uNKs [129], and the other primarily focusing on peripheral blood or conventional NKs that may not be influenced by the decidual cellular microenvironment to the same extent [127].

Consistent with the finding that maternal obesity results in hyper-activation of uNKs in mice, a recent study examining uNKs in obese women showed elevated cytotoxic granular content and cell surface expression of the activating KIR2DS1 receptor in the first trimester of pregnancy [224]. This increase in KIR2DS1 promoted an increase in TNF-α production [224], indicating that obesity in some way programs uNKs to acquire pro-inflammatory characteristics that may initiate or drive cellular changes within the uterus. While TNF-α contributes to normal spiral artery remodeling through promoting vascular smooth muscle cell apoptosis [225], it may also play roles in impairing placental development by reducing trophoblast survival and migration [47]. A way to interpret these findings relates to the paradigm that over- and under-activation of uNKs are detrimental to overall pregnancy health, where appropriate activation necessitates adequate engagement of activating receptors (i.e., KIRSDS1, NKp46) to promote processes like uterine arterial remodeling [226,227]. For example, in experimental circumstances where uNKs acquire hyper-activated states induced via alloantigen engagement [228] or by exposure to bacterial endotoxin [142,229], both placental insufficiency and fetal demise are observed. At the other end of the spectrum, in vitro work using human uNKs suggests that the presence of a dominant inactivating NKR, like KIR2DL1, may actually prevent uNK engagement needed for healthy placentation and pregnancy [226]. However, it is poorly understood how uNK activity may be dysregulated in adverse pregnancy conditions and this will be an important area of future study.

The impact of obesity in potentiating Mφ and T cell activation is well documented [230,231,232]. However, information on how maternal obesity shapes uterine Mφ and T cells is scarce. One study showed reduced frequencies in CD4^+^ and CD8^+^ T cells within the uterine tissues of DIO mice [127]. The impact of obesity on placental Mφ functions has also been examined [176,233]. In term placentas of obese women, placental Mφ was shown to adopt a pro-inflammatory state characterized by elevated expression of inflammatory mediators [176]. Similarly, in mice exposed to an obesogenic diet during pregnancy, placental Mφ showed elevated expression of inflammatory factors such as *Mcp-1*, colony stimulating factor (*Csf*)-1, colony stimulating factor receptor (*Csfr)2*, and a modest reduction in *Arg1* expression, which is highly expressed by immunoregulatory Mφ [233]. On the maternal side, pregnancy conditions associated with aberrant inflammation (i.e., pre-eclampsia, spontaneous preterm labor) also show alterations in Mφ biology, where decidual Mφ adopt features consistent with an M1-like phenotype [234,235,236]. In obesity, a strict M1-like phenotype in decidual Mφ has not been observed, but maternal obesity does associate with reduced numbers of decidual Mφ having an M2-like state [237]. Importantly, the approach used to immunophenotype Mφ in this latter study did not consider all Mφ populations, and therefore, a level of caution is warranted in interpreting these findings. However, in the rat, obesogenic diet exposure led to an overall increase in decidual Mφ, potentially driven by heightened levels of MCP-1 [130].

Impairments in trophoblast invasion associate with alterations in the ratio of decidual M1/M2 Mφ. Additionally, imbalances in both pro-inflammatory and regulatory decidual Mφ populations associate with multiple pregnancy disorders [234,235,236,238]. For instance, coculture of trophoblast cells with Mφ activated through exposure to inflammatory stimulus (e.g., LPS, TNF-α) coincided with restricted trophoblast invasion and trophoblast survival [239,240,241]. Similarly, treatment of first trimester placental villous explants cultures with recombinant TNF-α impaired trophoblast invasion [47]. Altogether, these findings suggest that Mφ activated through inflammatory cues may indirectly impair artery remodeling by limiting trophoblast invasion into uteroplacental arteries. However, a limitation of these studies is the reliance on peripheral blood monocytes used to derive Mφ instead of using true decidual Mφ. Clearly, the plasticity of Mφ and the challenge in culturing/maintaining tissue-associated Mφ will likely necessitate tempering of all in vitro-derived Mφ findings. Future studies deciphering the effects of maternal obesity on decidual Mφ would be of critical importance to understand the underlying decidual Mφ biology leading to pregnancy complications. Overall, these studies indicate that maternal obesity-linked inflammation promotes activation of uNKs and likely decidual Mφ as well. Two possible scenarios may occur in the uterine environment: (a) obesity induces a chronic state of uNK and decidual Mφ activation, leading to immune cell exhaustion and sub-optimal function, or (b) obesity’s impact on impaired or delayed utero-placental remodeling causes a compensatory response through uNK and decidual Mφ activation. 

*Maternal obesity-driven activation & exhaustion of uNKs?* NKs within the adipose tissue can become exhausted after persistent antigen presentation much faster in obese than in lean subjects [220]. It is possible that uNKs showing elevated activation in obese women might also experience exhaustion at faster rates than uNKs from healthy weight women resulting in inadequate uNK responses. For example, impaired uNK responses to HLA-C and HLA-G expressing invasive trophoblasts may result in insufficient uNK-mediated secretion of cytokines (e.g., TNF-α, IFN-γ) and angiogenic factors (e.g., VEGF, PLGF) necessary for vascular remodeling and decidual angiogenesis. However, this possibility has not yet been investigated, and further studies need to be conducted to examine if uNKs acquire an exhaustion-like phenotypes (e.g., expression of PD-1, Tim3, TIGIT), and whether these features associate with and contribute to negative pregnancy outcomes.

*Maternal obesity drives a compensatory response in uterine leukocytes?* While uterine vascular impairments are seen at mid-pregnancy in DIO rodent models, it is notable that pregnancy outcomes in these mice (i.e., fetal loss, fetal weight, litter size) are comparable to control mice [129,130]. Importantly, in these models, assessment of uterine arterial remodeling in later pregnancy shows that remodeling is indistinguishable from control mice, suggesting that initial impairment or developmental delay in immune-cell-mediated artery remodeling has caught-up. Specifically, these findings indicate that obesity may lead to hyperactivation of uNKs (and likely decidual Mφ), which in turn coordinate this compensatory mechanism. This possibility is substantiated by the observation that uNKs from DIO mice show increased expression of the activating receptor natural cytotoxicity triggering receptor (NCR)1 [129], a cytotoxic receptor previously shown to be essential in mediating uterine artery remodeling [242]. Furthermore, findings in humans show that maternal obesity associates with heightened uNK degranulation, as assessed by surface CD107a expression and increased production of TNF-α [224], corroborating this association between obesity and uNK activity.

It remains unclear whether these cellular compensatory mechanisms would have an impact on maternal-fetal tolerance. While findings in DIO mice suggest that tolerance is not impacted, it may be that obesity-induced immune-alteration over-sensitizes uNKs to infection or conditions with underlying inflammation to surpass what is normally a tolerogenic threshold. Future research designed to test if indeed a two-hit-like hypothesis (i.e., obesity + infection) potentiates uNK cytotoxicity will be necessary to promote understanding of this phenomenon.

## 4. Proposed Model of Maternal Obesity in Early Pregnancy

Despite substantial evidence supporting the theory that persistent inflammation alters immune cell responses, not much is known about local responses to obesity-induced inflammation within the maternal-fetal interface. As discussed earlier, immune cells contribute to placentation and pregnancy success through controlling artery remodeling and promoting a healthy tolerogenic environment. Here, we propose a model (Figure 3) describing potential effects of maternal obesity-linked inflammation on the biology of diverse sets of immune cells and how these interactions may lead to poor outcomes in pregnancy.

## 5. Concluding Remarks

Maternal obesity contributes to many complications in pregnancy, likely because of low-grade inflammation impacting arterial remodeling, placentation, and uterine immune cell composition and activity. Around 13% of the world’s population is obese [248]; thus, it is crucial that knowledge of the mechanisms underlying obesity-induced pregnancy complications is expanded, so that future treatment options can be elucidated. Mechanisms that could be studied further include composition and activity of uterine immune cells in obesogenic pregnancies, and effects of low-grade inflammation on decidualization and uterine arterial remodeling, as well as placental development and invasion.

## Figures and Tables

**Figure 1 ijms-21-03776-f001:**
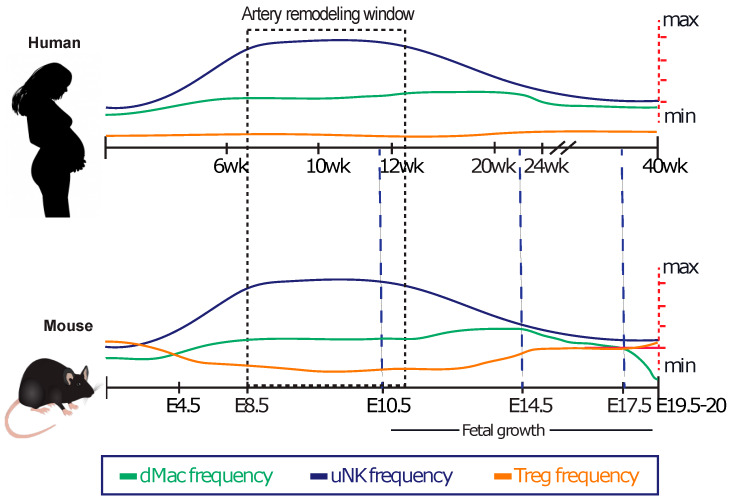
Timeline of human and mouse pregnancies with relative abundance of major decidual leukocytes. Important hallmarks during pregnancy are shown as well as the abundance of decidual Mφ (green), uNKs (blue), regulatory T cells (Tregs) (orange), and total CD3^+^ T cells (purple) [23,24,25]. Y-axis represents percent of total leukocytes in the decidua. wk, week; E, embryonic day.

**Figure 2 ijms-21-03776-f002:**
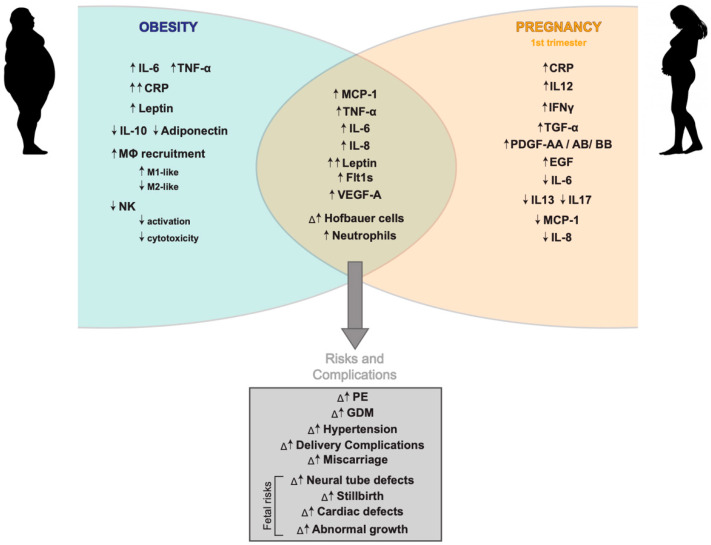
Schematic diagram of major inflammatory and immune characteristics observed in obesity and pregnancy. Important pro- and anti- inflammatory secretory factors released in the context of obesity and early pregnancy and their overlap during maternal obesity, along with the risks and complications this poses for both the fetus and the mother throughout pregnancy are shown.

**Figure 3 ijms-21-03776-f003:**
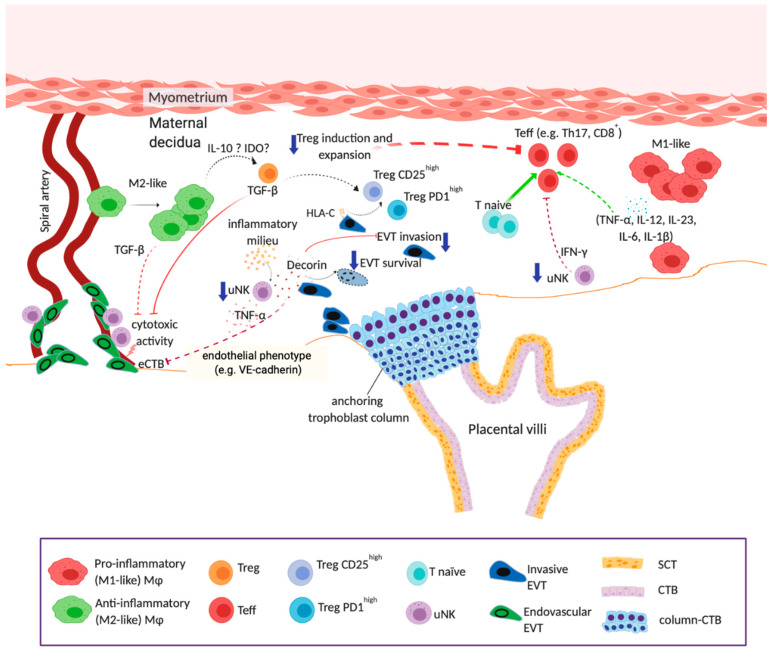
Proposed model of the effects of obesity on the intrauterine environment. Obesity-linked inflammation may affect uterine immune cell biology by: (**a**) disrupting the crosstalk between trophoblasts and uterine leukocytes resulting in impaired trophoblast invasion and artery remodeling [129,185,211,224,243,244,245], and (**b**) creating imbalances in immunoregulatory and pro-inflammatory uterine leukocytes that in turn potentiate maternal-fetal intolerance [60,211,246,247]. When inflammation is present in the intrauterine environment, homeostasis is disrupted leading to alterations in the frequencies and functions of important immunoregulatory cell populations that in turn impact spiral artery remodeling and trophoblast functions (invasion) and survival. Dashed lines indicate inefficient or inadequate induction/activation (depicted in green) or blockage/inhibition (depicted in red). Abbreviations: Teff, T effector cell; T naïve, naïve T cell; SCT, syncytiotrophoblast; CTB, villous cytotrophoblast; column-CTB, anchoring column cytotrophoblast. This figure was created using BioRender.com.

**Table 1 ijms-21-03776-t001:** Summarizes animal models of inflammation in pregnancy.

Model	Animal	Experimental Conditions	Results	Source
Obesity	Mouse	High-fat diet	∆ gut microbiota composition with ↓ gut barrier integrity ↓ artery remodeling and placentation ↑ placental inflammation, hypoxia, uNK cytotoxicity, fetal demise	[127,128,129]
	Rat	High-fat diet	↑ trophoblast invasion, MMP9 ↓ artery remodeling	[130]
	Rabbit	Cholesterol enriched diet	↑ concentration of total-cholesterol and lipoproteins ↓ fetal weight	[131]
Hypertension	Rat	Reduced uterine perfusion pressure procedure	↑ hypertension (mediated through Th cells and AT1) ↑ RORγ+CD4+ T cells and secretion of inflammatory factors	[132,133,134]
	Mouse	Angiotensin II infusion	↑ T cell composition in aorta, and aortic hypertrophy	[135]
Type 1 Diabetes	Mouse	Non-obese diabetic (NOD) mice	↑ placental weights, IFN-γ, tolerogenic antigen presenting cells ↓ uNK, vascular remodeling, fetal weights, T cells in offspring	[136,137,138,139]
Viral infection	Mouse	Allogeneically mated female injected poly I:C	↑ fetal loss, cytotoxicity of CD27^pos^DX5^pos^CD3^neg^ NK ↓ proportion of CD27^low^DX5^pos^CD3^neg^ NK	[140]
Sterile inflammation	Mouse	Administration of monosodium urate crystals during late gestation	↑ placental inflammation and FGR ↓ fetal weight = placental weight, placental morphology	[141]
Bacterial infection	Mouse	Mice were administered lipopolysaccharide (LPS) intraperitoneally (i.p.)	↑ uNK, % NK1.1+CD27+ uNK, and NK-mediated fetal demise ↑ pulmonary and placental cytokine production	[142,143]
	Rabbit	Transcervical *P. bivea* inoculation	↑ preterm delivery, chronic intrauterine, and fetal infection	[144]
FGR	Rat	4-day low-dose LPS injections i.p.	↓ placental development, vascular remodeling, fetal growth	[145,146,147,148,149]
		Chronic leptin treatment	↑ TNF-α expression in placenta ↓ maternal, fetal, and placental weights	[150]
	Canine	Maternal nutrient deprivation	↓ fetal weight, maternal blood glucose, ketone bodies, and FFA	[151]
Preterm labor	Rat	Intrauterine injury induced by transient hypoxia-ischemia and LPS injection	↑ acute and subacute placental injury ↑ inflammatory factors in the placenta	[152]
		i.p. injection of LPS	↓ birth weights (alleviated with erythromycin treatment)	[153,154]
	Mouse	i.p. injection of LPS	↑ TLR-4-mediated preterm birth with no neonatal mortality	[155,156,157,158,159,160]
		3 injections of rhIL-1	Parturition occurred within 24 h	[161]
		Lipoteichoic acid i.p.	↑ incidence of preterm delivery	[162]
	Rabbit	LPS administered via catheter to uteri five times at 1-h intervals	↑ uterine contractions ↑ prostaglandin production	[163]
		Endocervical *E. coli* inoculation	↑ Inflammation in uterus, placenta, and fetal lung	[164]
	Sheep	Intravenous LPS	↑ fetal hypoxemia ↑ fetal and maternal hypothalamo-hypophyseal-adrenal axis	[165,166]
	Monkey	*Streptococci* inoculation	↑ spontaneous parturition, steroid hormones	[167,168]
Genetic models of inflammation	Mouse	IFN-γ-/- or IFN-γRα^−/−^ mated with BALB/c male	↑ uNK, fluid, cellularity, and necrosis in decidua ↓ uNK granularity and decidual artery remodeling	[169]
		RAG-2-/-yc-/- or IL-15-/- pregnant females received bone marrow from IL-15-/- or C57/BL6 pregnant females IRF-1-/- females treated with mrIL-15 on GD5 for 5 days	RAG-2-/-yc-/- + bone marrow from IL-15-/- ↑ uNK, uNK differentiation, decidual artery remodeling RAG-2-/-yc-/-, or IL-15-/- + bone marrow from C57/BL6 ↓ uNK, MLAp, decidual cellularity and artery remodeling IRF1-/- + mrIL-15 ↓ uNK differentiation, artery remodeling, placental and fetal weight	[170]
		*Lif*-/- females mated with MF-1 males	↓ macrophages and NKs in mesometrial stroma ↑ eosinophils, NKs in uterus	[171]
		D8 male father (containing H2-D^d^ MHC allele)	↓ NK function, uterine arterial remodeling, fetal growth	[172]
		BALB/cx C57/BL6 (FxM) compared to C57/BL6xBALB/c	↑ decidual artery diameters	[173]

↑ indicates increase; ↓ indicates decrease in frequency, expression, and amount.

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
