# Peer review of "Maternal Obesity and the Uterine Immune Cell Landscape: The Shaping Role of Inflammation"

_ijms, 2020, doi:10.3390/ijms21113776_

Round 1
Reviewer 1 Report
St Germain et al present a well written, topical narrative review examining the links between uterine immune cells and maternal obesity.
Overall this is a comprehensive review with provides a detailed account of the role of uterine immune cells to pregnancy outcomes and examines the impact of maternal obesity in an inflammatory context. I only have a few very minor comments:
- The authors detail both human and rodent pregnancies. There are a range of differences in terms of both pregnancy endocrine regulation and placentation between human and rodents. It would be nice to have a short paragraph to detail these differences for clarity.
- While some of the cytokines/chemokines are discussed in this review, there are some key mediators which contribute to both the chronic low-grade inflammation which is typical of obesity induced metabolic dysfunction and adverse pregnancy outcomes. For examples IL-1 signalling has barely been mentioned. I think it would benefit the review to have a more detailed discussion of some of these key inflammatory mediators.
Author Response
We thank the reviewer for providing us insightful comments on our review manuscript. Below we have listed the changes we have made to the revised manuscript that we feel have strengthened its overall readability and quality.
1) There are a range of differences in terms of both pregnancy endocrine regulation and placentation between human and rodents. It would be nice to have a short paragraph to detail these differences for clarity.
We agree with the reviewer that the original manuscript did not do an adequate job in describing the differences between human and rodent features of pregnancy. To this end, we have addressed this by including a new paragraph on page 10 Lines 324-338 that details human-mouse differences in placental structure, trophoblast biology, and endocrine production that may interact with inflammatory responses in pregnancy.
2) While some of the cytokines/chemokines are discussed in this review, there are some key mediators which contribute to both the chronic low-grade inflammation which is typical of obesity induced metabolic dysfunction and adverse pregnancy outcomes. For examples IL-1 signalling has barely been mentioned. I think it would benefit the review to have a more detailed discussion of some of these key inflammatory mediators.
We thank the reviewer for identifying this important omission, and we have added new text on page 7 Lines 251-255 that briefly highlight key obesogenic factors like IL1-beta and HMGB1 that contribute to aberrant inflammation.
Reviewer 2 Report
This is a review article on the effects of maternal obesity on immune cells and inflammation, with specific application to endometrium/decidua and pregnancy complications. The subject is new, interesting and of great potential scientific and clinical relevance. The manuscript is well designed, well structured and well written. The effort and the success of the authors in disentangleing this extremely complex matter is evident.
I have few minor suggestions.
Whenever possible, the authors compared the results obtained in mice with those obtained in humans. This is a great merit of the article, because it is actually difficult to realize a beautiful synthesis. The authors were successful in obtaining this. However, there is also the need to remark the differences between humans and mice, particularly in their specific placentas (Arck PC, Hecher K. Nat Med 2013, 19, 548-56).
- lines 139-141: it could be better use the term "fetal semiallograft". The role of endometrial/decidual immune cells in recurrent pregnancy loss is much more complex than that reported in the sentence (Ticconi C., et al. Int J Mol Sci 2019 oct 26). This should be made clear.
The authors may consider to add few/several clinical aspects to their consideratons. For instance, after line 217 some interesting clinical observations could be added (Bhandari HM et al., BJOG 2016, 132, 217-22.; Metwally M, et al. Fertil Steril 90:714-26, 2008; Metwally M, et al. Fertil Steril 94: 290-5, 2010).
Fig. 2: PE, GDM, hypertension and delivery complications are disorders of late pregnancy. This could be clarified in the legend, since the title of the Fig. is referred to first trimester.
Did the authors consider a possible role of HMGB1 as a mediator of the obesity-driven pregnancy complications in the setting of inflammation?
Author Response
We thank the reviewer for providing us with insightful comments. We have addressed each comment, and listed below we detail specifics on what aspects have been changed. We feel these new changes have improved the manuscript's overall quality.
1) However, there is also the need to remark the differences between humans and mice, particularly in their specific placentas (Arck PC, Hecher K. Nat Med 2013, 19, 548-56).
We thank the reviewer for identifying this gap in our initial submission. We have now included a new paragraph on page 10 Lines 324-338 that specifically addresses key differences between the human and mouse placenta.
2) lines 140-142: it could be better use the term "fetal semiallograft". The role of endometrial/decidual immune cells in recurrent pregnancy loss is much more complex than that reported in the sentence (Ticconi C., et al. Int J Mol Sci 2019 oct 26). This should be made clear.
We apologise for our initial oversight in inappropriately referring to the fetus as an allograft. We have modified the language to reflect that it is indeed a semiallograft (Page 4, lie 141). We have additionally expanded this sentence to acknowledge that the role of immune cells in recurrent pregnancy loss is complex (Page 4, lines 140-142).
3) The authors may consider to add few/several clinical aspects to their considerations. For instance, after line 217 some interesting clinical observations could be added (Bhandari HM et al., BJOG 2016, 132, 217-22.; Metwally M, et al. Fertil Steril 90:714-26, 2008; Metwally M, et al. Fertil Steril 94: 290-5, 2010).
We thank the reviewer for this suggestion. As indicated, we have now added new text (Page 6 Line 233) and appropriate references related to obesity-associated pregnancy outcomes.
4) Fig. 2: PE, GDM, hypertension and delivery complications are disorders of late pregnancy. This could be clarified in the legend, since the title of the Fig. is referred to first trimester.
We thank the reviewer for this initial oversight. We have modified the figure legend text to now reflect obesity-associated pregnancy complication across gestation.
5) Did the authors consider a possible role of HMGB1 as a mediator of the obesity-driven pregnancy complications in the setting of inflammation?
We have now included a brief statement (Page 7, Lines 251-255) that refers to HMGB1 and IL1-beta as mediators of obesity-associated inflammation on driving poor pregnancy outcomes.
Reviewer 3 Report
This is a very good review on the local uterine immune mechanisms in obese pregnant women, a difficult topic in human physiology of which current knowledge is rather poor. The paper is well written, the information complete and update, the references adequate. The presented data clearly make the case that maternal local inflammation processes differ between obese and lean women. There are a few suggestions that may help the authors to make this case even stronger.
- Today, basically two different types of preeclampsia are reported, most commonly labelled as early onset and late onset preeclampsia. The early onset type is characterised with first trimester signs of endothelial dysfunction and activated state of systemic inflammation. Late onset preeclampsia initially starts off normally and converts to an enhanced state of endothelial dysfuntion during the course of pregnancy. Late onset preeclampsia is highly represented by obese women. The presumed decidual processes in these two different types of preeclampsia is discussed in a paper by Redman C “Redman CW, Sargent IL, Staff AC. IFPA Senior Award Lecture: making sense of pre-eclampsia - two placental causes of preeclampsia? Placenta. 2014 Feb;35 Suppl:S20-5. doi: 10.1016/j.placenta.2013.12.008. Epub 2014 Jan 11. PubMed PMID: 24477207”. Do different local uterine immune mechanisms predispose to either early and late onset preeclampsia ?
- Next to an altered immune function in obese versus lean women, other physiological parameters also differ such as circulating volume & cardiac output, autonomic nervous activity, intra-abdominal pressure, … not to mention the endocrine/metabolic differences. An important point of this is that these differences are present already before conception. Can the authors comment on how the suboptimal local uterine immune activity in obese women depends on this pregestational state and/or on an inadaquate gestation induced adaptation ?
- In line with the two former remarks, can the authors comment on the immunologic fetal-maternal communication mechanisms, expanding the abnormal local uterine immune response to a state of generalised systemic inflammation ? There is histologic evidence that trophoblast invasion of local lymphatic capillaries occurs before the remodelling of spiral arteries. And there is also evidence that the immune system is strongly involved in fetal-maternal communication throughout the course of pregnancy.
- Can the authors comment of different local immune responses between primi- and multigravid women and in women with preexisting auto-immune diseases such as antiphospholipid syndrome? Is obesity also associated with an altered immune response here ?
As a whole, I think this is a marvelous paper for which I sincerely congratulate the authors.
Author Response
We thank the review for the critical read of our manuscript and the suggestions and comments that we have addressed (detailed below) and we feel have strengthened its overall clarity and quality.
1) Do different local uterine immune mechanisms predispose to either early and late onset preeclampsia?
We thank the reviewer for raising this interesting point related to differences between early and late onset preeclampsia, and how obesity and inflammation may contribute or predispose to the development of these disorders. We have added new text (Page 7, Lines 255-259) that acknowledges the association between inflammation and preeclampsia, the underlying roles of aberrant placental function (early-onset) and existing maternal syndromic factors (late onset) that may contribute to the development of these preeclampsia subtypes. Additionally, we provide a statement that associates late onset preeclampsia more frequently with maternal obesity, and speculate that alterations in inflammation and metabolism may contribute to this preeclampsia subtype.
2) Can the authors comment on how the suboptimal local uterine immune activity in obese women depends on this pregestational state and/or on an inadaquate gestation induced adaptation?
We thank the reviewer for suggesting this important point. We have now included new text (Page 7, Lines 260-264) that addresses the importance of pre-existing health conditions associated with obesity and how these may influence altered uterine immune function in pregnancy.
3) In line with the two former remarks, can the authors comment on the immunologic fetal-maternal communication mechanisms, expanding the abnormal local uterine immune response to a state of generalised systemic inflammation ? There is histologic evidence that trophoblast invasion of local lymphatic capillaries occurs before the remodelling of spiral arteries. And there is also evidence that the immune system is strongly involved in fetal-maternal communication throughout the course of pregnancy.
We have added new text that expands on how trophoblast functions and trophoblast-decidual interactions may be impacted by inflammation/obesity (Page 10, Lines 334-337; Pages 10 & 11, Lines 346-367).
4) Can the authors comment of different local immune responses between primi- and multigravid women and in women with preexisting auto-immune diseases such as antiphospholipid syndrome? Is obesity also associated with an altered immune response here ?
We thank the review for suggesting the addition of these important concepts. We have added new text to Page 7, Lines 260-264 that outline how preexisting maternal health and prior exposure to paternal antigen may affect local uterine immune cell responses.
5) As a whole, I think this is a marvelous paper for which I sincerely congratulate the authors.
We thank the reviewer for their support and constructive critique.